# A Single Standard to Determine Multi-Components Method Coupled with Chemometric Methods for the Quantification, Evaluation and Classification of *Notopterygii* Rhizoma et Radix from Different Regions

**DOI:** 10.3390/molecules24193574

**Published:** 2019-10-03

**Authors:** Xie-An Yu, Jin Li, John Teye Azietaku, Wei Liu, Jun He, Yan-Xu Chang

**Affiliations:** 1Tianjin State Key Laboratory of Modern Chinese Medicine, Tianjin University of Traditional Chinese Medicine, Tianjin 300193, China; yuxieanalj@126.com (X.-A.Y.); Lijin@tjutcm.edu.cn (J.L.); alancogee@yahoo.com (J.T.A.); liuweict@sina.com (W.L.); 2Tianjin Key Laboratory of Phytochemistry and Pharmaceutical Analysis, Tianjin University of Traditional Chinese Medicine, Tianjin 300193, China

**Keywords:** chemometrics, different regions, *Notopterygii* rhizoma and radix

## Abstract

An ultra-high-performance liquid chromatography-quadrupole/time of flight mass spectrometry is used to identify 33 compounds in *Notopterygii* rhizoma and radix, after which a single standard to determine multi-components method is established for the simultaneous determination of 19 compounds in *Notopterygii* rhizoma and radix using chlorogenic acid and notopterol as the internal standard. To screen the potential chemical markers among *Notopterygii* rhizoma and radix planted in its natural germination area and in others, the quantitative data of 19 compounds are analyzed via partial least-squares discriminant analysis (PLS–DA). Depending on the variable importance parameters (VIP) value of PLS–DA, six compounds are selected to be the potential chemical markers for the discrimination of *Notopterygii* rhizoma and radix planted in the different regions. Furthermore, the Fisher’s discriminant analysis is used to build the models that are used to classify *Notopterygii* rhizoma and radix from the different regions based on the six chemical markers. Experimental results indicate that *Notopterygii* rhizoma and radix planted in the Sichuan province are distinguished successfully from those in other regions, reaching a 96.0% accuracy rating. Therefore, a single standard to determine multi-components method combined with a chemometrics method, which contains the advantages such as simple, rapid, economical and accurate identification, offers a new perspective for the quantification, evaluation and classification of *Notopterygii* rhizoma and radix from the different regions.

## 1. Introduction

*Notopterygii* rhizoma and Radix (NRR), ‘qianghuo’ in Chinese, is a traditional Chinese medicine (TCM) planted in the Sichuan, Gansu, Tibet and Qinghai provinces of China. NRR, the rhizomes and roots of *Notopterygium incisum* Tingex H.T. Chang (NI) and *Notopterygium forbesii* Boiss (NF) [1] has been used as a famous antirheumatic and analgesic medicine for curing rheumatism, headache, cold, neuralgia and arthralgia [2,3]. Moreover, NRR has been used as an essential ingredient in a large number of Chinese prescriptions for the treatment of diseases since Ming dynasty. Since it is planted in different geographic regions, the composition and the clinical efficiency of NRR can demonstrate significant difference. The original producing area of NRR is Sichuan province, ‘Chuan qianghuo’, whose product is valued highly for its superior quality. The NRR planted in different regions are similar in morphological shape and difficult to distinguish accurately by the naked eye, leading to market confusion regarding NRR. Therefore, it is essential to develop a simple, rapid, economical and accurate method to evaluate and distinguish the NRR planted in Sichuan or other areas. Currently, some methods developed for the quality control of NRR, include HPLC [4,5,6], GC–MS and HPLC–DAD–ESI–MS [7]. To the best of our knowledge, however, these still are difficult for the determination of several compounds in the NRR to allow us to comprehensively evaluate and classify the different geographical regions of NRR production.

A single standard to determine multi-components (SSDM) method has been employed in quality evaluations of Chinese herbal medicine and is becoming a novel and efficient technique [8,9,10,11,12]. One standard is used to determine several components with similar structure, which offers an economical method for the simultaneous determination of multi-components in Chinese herbal medicine [13,14,15,16,17]. Facing the current bottleneck of the scarcity and high price of standards, a SSDM method also has been tested and verified as an alternative and promising method to comprehensively and effectively control the quality of Chinese herbal medicines. Moreover, Chinese herbal medicines planted in different regions would increase the differences of chemical components and, therefore, is still an obstacle to conducting a synthetical evaluation and classification of the area of origin (Sichuan) products just according to the simultaneous determination of major compounds. 

Chemometrics is a discipline related to the application of mathematics, statistics and computer science [18,19]. The aim of the chemometrics is to analyze the data of the multidimensional chemical information, design and choose the appropriate algorithm to screen the main chemical information in the complex samples which are suitable for the comprehensive profiling and characterization of the multicomponents from Chinese herbal medicine [20,21]. Feng et al. developed the UHPLC–QTOF/MS and HPLC–PDA with chemometric methods to explore global chemical profiling and discriminate the *Chimonanthus nitens* Oliv. leaf (CNOL) from different geographical origins [20]. Kuang and colleagues constructed the UPLC–ESI^−^–MRM–MS with chemometric methods to evaluate American ginseng cultivated in the Heilongjiang province of China [22]. Wang and co-workers also established the UPLC–TQ–MS with multivariate statistical analysis methods to investigate an efficient analytical procedure for the QC of licorice [23]. The integration of chromatographic analysis and multivariate statistical analysis, with the development of chemometrics, would be a credible tool to investigate a plant exploited in Chinese herb medicine. Hence, the chemometric methods can be used as an effective tool to explore the difference of Chinese herb medicine planted in different geographical origins, which is an excellent multivariate statistical method to identify and classify the similarities or differences of Chinese herb medicine based on the multi-compounds.

During this study, an UHPLC–Q–TOF/MS method is used to identify 33 compounds in NRR followed by a novel SSDM method to determine 19 major compounds in NRR. Coumarins and organic acids are the main compounds in NRR [3]. Hence, chlorogenic acid and notopterol are chosen as the internal standard for determining the 19 compounds from NRR. Next, according to the quantitative data of 19 compounds, a partial least-squares discriminant analysis (PLS–DA) is used to select the potential markers that cause the chemical difference of NRR planted in different regions. As a result, six compounds, including Nodakenin, Diosmetin, Imperatorin, Osthole, Notopterol and Praeruptorin A are selected successfully, which can be used to establish the standard models to evaluate and classify NRR from different regions via Fisher’s discriminant analysis (FDA). The NRR planted in Sichuan and other regions are classified, with a 96.0% correctness in their classification. Overall, the proposed method can be used for the determination of the multi-compounds, quality control, and evaluation of the different regions of NRR and other Chinese herb medicines.

## 2. Result and Discussion

### 2.1. Optimization of Extraction Conditions

To achieve an efficient extraction of the 19 compounds in NRR, the main parameters, including extraction solvent, sample/solvent ratio and extraction time, were optimized. The single factors tests first were employed to obtain a reasonable range of data for the following orthogonal experiment. The dried powdered NRR samples were weighed at 1.0 g and then were extracted ultrasonically with a 10 mL methanol-water solution (40%, 60%, 80% and 100%) for 30 min. The 100% methanol-water solution produced a higher extraction efficiency. Different rations of the sample/solvent (1.0 g/5 mL, 1.0 g/10 mL and 1.0 g/20 mL) were investigated to obtain the ideal extraction yield. The results showed that the sample/solvent of 1.0 g/10 mL can reach a higher extraction yield. A 1.0 g sample was added to 10 mL methanol and extracted with ultrasonics for 20 min, 30 min, 40 min and 60 min. The extraction yield was significantly increased with the extraction time increasing from 20 to 30 min. The total contents of the 19 compounds increased minimally within 30–60 min. Hence, a 30 min extraction time was chosen for further experimentation. According to these results of the single factors tests, the extraction solvent (methanol concentration 60%, 80% and 100% *v*/*v*), sample/solvent ratio (1:5, 1:10 and 1:20 *w*/*v*) and the ultrasonication time (20, 30, 40 min) were optimized using an orthogonal L9 (3^4^) experiment. The optimum sample extraction condition was achieved at sample/solvent ratio 1:10, 100% *v*/*v* methanol and ultrasonication for 30 min. 

### 2.2. Optimization of Chromatographic Conditions 

To ensure the 19 active compounds were well separated, the chromatographic conditions, including mobile phase composition, addition of formic acid concentrations in water phase, column temperature, and the flow rate, were optimized. Acetonitrile, methanol and the mixed solvent with different percentages for the organic phase and different formic acid concentrations (0.05%, 0.1% and 0.2%) for the water phase were optimized. Different column temperatures (25, 30, 35 and 40 °C) and flow rates (0.2, 0.3 and 0.4 mL min^−1^) also were tested. Consequently, the acetonitrile and methanol, mixed at a ratio of 1:1 and 0.1% formic aqueous solution, were optimized as the mobile phase. The flow rate was 0.3 mL min^−1^ and the column temperature maintained at 35 °C. The wavelength was set at 325 nm to detect both coumarins and organic acids in NRR. The chromatograms of the *Notopterygii* rhizoma and radix sample are shown in Figure 1. 

### 2.3. Calculation of the Relative Response Factors

The first step in calculating the relative response factors was choosing the internal standards. Taking important note of the characteristics of the compound into consideration, including the prices, the content and stability, chlorogenic acid and notopterol were chosen to be the internal standard for 4 organic acids and 15 coumarins, respectively. Then, the relative response factors (R_f_) were calculated using the formula in Section 3.4. The details of the relative response factors of all compounds are shown in Table 1.

### 2.4. Method Validation

A calibration curve was built by plotting the peak areas of each analyte versus its concentrations using eight concentration levels of the 19 compounds. The data displayed in Table 1 for the 19 compounds had a good linear relationship (r > 0.9996) within their range. The limit of detection (LOD) and limit of quantification (LOQ) were calculated by diluting the concentration of the mixed standard until reaching the signal-to-noise (S/N) ratio of 3 and 10 in the chromatography, respectively. The range of LOD for the 19 compounds was 0.02–0.15 μg mL^−1^, and the range of LOQ was 0.08–0.4 μg mL^−1^. The details of each compound are shown in Table 1.

Testing for precision involved both intra-day and inter-day variabilities. Three concentrations of the QC sample in six replicates were analyzed for precision. According to the detail results in Table 2, the inter-day precision showed that the accuracy range for SSMD was 85.7%–114%, and the RSD was less than 6.32% for SSMD. Found when testing for intra-day precision, the accuracy ranged from 84.7%–114% for SSMD, with an RSD less than 5.09%. 

All of the 19 compounds were found to be stable at room temperature until 24 h; the result in Table 3 shows that three different concentrations of QC samples were stable with the range of 83.2–121% using SSMD methods and the RSD was less than 6.51%.

The accuracy of recovery was measured by spiking three different concentrations to nearly 80%, 100% and 120% of the original sample content. Regarding the SSMD method, the accuracy was in the range of 82.9–123%, and the RSD was within 8.10%, except for Diosmetin. The concentration of Diosmetin was only 0.04 μg mL^−1^, which was below the LOQ, so the accuracy range was out of the regular range. The result is shown in Table 4. It was shown that the SSDM method was accurate within the linear range. Conversely, if the analysis concentration is out of the range, the SSDM method may not be suitable for the determination.

To compare the method validation using the SSDM and external standard (ES) methods, the method validation was also carried out using ES methods, the results are shown in the Supporting documents (Appendix A).

### 2.5. Identification of 33 Compounds in NRR Using UHPLC–Q–TOF–MS 

During the study, both negative and positive modes were used to identify the 33 compounds in NRR. As shown in Table 5 and Figure 2, 13 phenolic acids and 20 coumarins were identified. Regarding the negative mass spectra, peak 1 had a molecular weight of 116.0036 with ions of *m*/*z* 115.0034 (C_4_H_4_O_4_) [M − H]^−^, while in the MS/MS spectra, the product ions were 53.0038 [M − H − H_2_O − CO_2_]^−^. Accordingly, comparing this result with the literature, peak 1 was identified as fumaric acid [24]. Peak 2, 5 and 6 exhibited the same parent ion [M − H]^−^ at *m*/*z* 353.0892, with the special fragment ions at 191.0559 [M − H − C_6_H_12_O_6_ + H_2_O]^−^ and 135.045 [M − H − C_6_H_12_O_6_ + H_2_O −2CO_2_]^−^ indicating the structure of chlorogenic acid [25]. They were identified as neochlorogenic acid, chlorogenic acid and cryptochlorogenin acid, respectively, with the additional information of retention times. Peak 7, 11 and 12 were also isomers with ions of *m*/*z* of 515.1196 (C_25_H_24_O_12_) [M − H]^−^; the fragment ions were 353.0916, 191.0558 and 179.0349. Taking their different retention times into consideration, they were identified as isochlorogenic acid B, isochlorogenic acid A and isochlorogenic acid C [25]. Peak 3 had the major first-order mass spectrum at *m*/*z* 341.0885 (C_15_H_18_O_9_) [M − H]^−^, and the MS/MS fragments 179.0347 [M − H − C_6_H_12_O_6_ + H_2_O]^−^ which could be a result of the glucoside expulsion from the parent ion, so it could be identified as Caffeic acid 3-glucoside [25]. Peak 4 had the mass spectrum at *m*/*z* 153.0186 (C_7_H_6_O_4_) [M − H]^−^, the product ion was 109.0287 [M − H − CO_2_]^−^, which could be identified as 2, 5-Dihydroxybenzoic acid. Peak 8 had the mass spectrum at *m*/*z* 163.0397 (C_9_H_8_O_3_) [M − H]^−^, the product ions were 119.0495 [M − H − CO_2_]^−^, therefore, it could be identified as p-coumaric acid [26]. Peak 9 had the parent ion at *m*/*z* 193.0507 (C_10_H_10_O_4_) [M − H]^−^, the target product ions 135.0411 and 89.0324 can be obtained in the MS/MS spectra, which could be used to identify peak 9 as ferulic acid [6]. Peak 10 showed the ion with *m*/*z* of 453.1405 (C_20_H_24_O_9_) [M − H]^−^, and the reference standard was of the same retention time, the fragment ions including 227.0703 and 89.0238, thus it can be treated as nodakenin [5].

Considering the positive mass spectra, the 20 compounds were mostly coumarins. Regarding peak 14, the parent ion had *m*/*z* of 163.0389 (C_9_H_6_O_3_) [M + H]^+^, and the product ion was 145.0482 [M + H − H_2_O]^+^. This, therefore, was identified as umbelliferone [27]. Peak 15 exhibited an ion with *m*/*z* 193.0492 (C_10_H_8_O_4_) [M + H]^+^, product ions included 150.0289 [M + H − CO_2_]^+^ and 122.0363 [M + H − CO_2_ − CO]^+^, thereby giving information that the peak was scopoletin [28,29]. Peak 16 had the MS spectra show a parent ion [M + H]^+^ at 207.0642, which formula was C_11_H_10_O_4_. The product ion was 146.0484, the structure was of coumarin [3], so this could be identified as 6,7-Dimethoxy coumarin. Peak 17 had a parent ion with molecular formula (C_11_H_6_O_4_) [M + H]^+^ and the fragment (147.0443) was obtained by loss of a 2 CO mass unit, hence, it was identified as bergaptol [28]. Regarding peaks 18 and 19, they were isomers with the parent ion having a *m*/*z* 187.0391(C_11_H_6_O_3_) [M + H]^+^ where the fragment was produced by a loss of 2 CO to give 131.0489. They finally were identified as Psoralen and Angelicin [30]. Another three types of isomers, peaks 22 and 25, peaks 28 and 32, and peaks 21 and 23, were shown to have parent ions with *m*/*z* of 247.0602 (C_13_H_10_O_5_) [M + H]^+^, *m*/*z* 271.0976 (C_16_H_14_O_4_) [M + H]^+^ and *m*/*z* 217.0510 (C_12_H_8_O_4_) [M + H]^+^, respectively. Looking at the fragment, 217.0134 [M + H − CO − 2H]^+^ and 189.0178 [M + H − 3CO − 2H]^+^ were produced by losing CO. Therefore, peaks 22 and 25 were identified as pimpinellin and isopimpinellin [31]. Concerning peaks 21 and 23, the product ion was 146.0453 [M + H − 2CO − CH_3_]^+^. They were identified as xanthotoxin and bergapten. Product ions with *m*/*z* 203.0382 [M + H − C_5_H_8_]^+^, 147.0433 [M + H − C_5_H_8_ − 2CO]^+^ were special ions, which could be used to identify peaks 28 and 32 as imperatorin and isoimperatorin [31]. Product ions at *m*/*z* 146.0453 [M + H − CH_3_ − 2CO]^+^ were obtained in the MS/MS spectra, hence, the peaks 21 and 23 could be identified as xanthotoxin and bergapten [30]. Peak 24 showed a parent ion at *m*/*z* 301.0721 (C_16_H_12_O_6_) [M + H]^+^, while the product ion was 217.0134 [M + H − 2CO − 2CH_2_]^+^. It was identified as diosmetin. Peak 30 had the MS spectra show a parent ion [M + H]^+^ at 245.1190, which formula was C_15_H_16_O_3_. The product ion was 189.1097 and 131.0491. Therefore, it could be identified as osthole [3]. Peak 31 showed a parent ion at *m*/*z* 377.1369 (C_21_H_22_O_5_) [M + Na]^+^, the product ion 203.0345 [M + Na-C_4_H_8_ − CO − C_5_H_8_]^+^, thus, the ion was comparable to imperatorin. It could be identified as notopterol [30]. The detail information is shown in Table 5.

### 2.6. Quantitative Analysis of 32 Batches of Samples

Once the method was validated successfully, 32 different samples were analyzed. Both ES and SSMD methods were used in the determination of the 19 active compounds, with the results shown in Table 6 and the Supporting documents (Table 5). To make the SSMD method more reliable, the Relative Deviation (RD) was calculated by the following formula:RD = (CSSMD − CES)/CES

The CSSMD and CES are the content of the 19 compounds by SSMD method and ES method.

The detailed RDs are shown in Table 7. The RDs were below 10%, indicating that the ES and SSMD methods were almost the same. According to the Chinese Pharmacopoeia (2015), the contents of notopterol and isoimperatorin should be more than 0.4% (4 mg g^−1^). Seven samples out of them were treated as irregular samples using both ES and SSMD methods.

### 2.7. Multivariate Statistical Analysis

As we all know, NRRs planted in the Sichuan province were treated as the superior quality “genuine” regional herb. Here, 32 batches, including 18 batches planted in the Sichuan province and 14 batches planted in other regions, were analyzed. According to the Chinese Pharmacopoeia (2015) rules that the content of notopterol and isoimperatorin in NRR should be more than 0.4%, there were 7 samples, including those from S1 (Sichuan Chengdu), S20 (Sichuan), S21 (Sichuan), S22 (Shandong Taishan), S28 (Sichuan), S29 (Tianjin) and S30 (Shanxi), tested and found to be irregular. Therefore, they were excluded from further analysis. To screen the potential compounds that caused the chemical difference between NRRs planted in Sichuan province and others, 25 batches of samples were subjected to PLS–DA to validate whether they could be grouped into two parts according to the 19 compounds. The 19 variables were autofitted for the model using cross validation rules to determine the number of significant components. The autofit model was the 5 latent variables, which can reflect 95% of the information from the 19 variables. Hence, the 5 latent variables were extracted to build the PLS–DA model. As shown in Figure 3A, the PLS–DA scores plot were classified readily into two zones with R2X = 0.949, R2Y = 0.800 and Q2 = 0.514, revealing a good classification ability of the model. Next, depending on the variable importance parameters (VIP > 1), 6 markers, including Nodakenin, Diosmetin, Imperatorin, Osthole, Notopterol and Praeruptorin A were selected and considered the key compounds, causing the chemical differences among different regions. Following that, the PLS–DA model was applied again to validate whether NRRs could be separated according to only the 6 markers. Similar to the PLS–DA scores plot according to the 19 compounds, the NRR samples from different regions also were separated into two zones (Figure 3C) with R2X = 0.966, R2Y = 0.763 and Q2 = 0.616, depending on the 6 markers. To validate the models, the model validity of the PLS–DA was checked by a permutation test with 200 interactions. The criteria for validity are shown in Figure 3B,D, with all red R2-values and blue Q2-values to the left lower than those on the right, suggesting that the established discriminant model was stable and reliable. Therefore, this demonstrated that the developed PLS–DA method was a powerful tool for the classification of the geographical origins of NRR. Subsequent to classification, the Fisher’s discriminant analysis (FDA) was used to build the predictive model for unknown samples based on the selected 6 markers. It produced a discriminant function based on the variables that provided the discrimination among the different regions of NRR. The discriminant functions were generated by the samples with known groups. Then, discriminant functions were used to predict the predictor variables with unknown groups. An unknown sample also was predicted as known group members (Sichuan or other regions). The discriminant analysis equation is expressed as: Y = A + b1x1 + b2x2 + b3x3 + b4x4 + b5x5 + b6x6 (Y is the discriminant function, b is the discriminant coefficients, x is the score of the explanatory variable, A is the constant). During our experiment, the discriminant analysis function was built by SPSS 25.0 and the function was expressed as: Y = −7.889 − 0.020x1 + 2.585x2 + 0.140x3 + 2.198x4 + 0.938x5 × 0.408x6, while x1, x2, x3, x4, x5, x6 stood for the constants of Nodakenin, Diosmetin, Imperatorin, Osthole, Notopterol and Praeruptorin A. The classification result showed that 96.0% of the originally grouped cases were classified correctly, and 92.0% of the cross-validation grouped cases were classified correctly. The average value was calculated to be −0.208, depending on the group 1 value of 1.526 and group 2 value of −1.943. This means that, using the discriminant analysis function, the unknown sample with the calculated value of more than −0.208 can be treated as the NRR planted in the Sichuan province (origin area medicine) or otherwise from the other regions. Viewing the result shown in Table 6, NRR samples produced in the Sichuan province can discriminate group 1 successfully, and other regions can be classified as group 2, except for sample 9 produced in Gansu province The Sichuan and Gansu provinces are geographically close in China, therefore, it is a reasonable phenomenon that some samples planted in Gansu would be similar to those planted in the Sichuan province. Meanwhile, the results provide strong evidence for the use of a discriminant analysis function as an effective tool to discriminate the different regions of Chinese herb medicines.

## 3. Materials and Methods

### 3.1. Chemicals and Materials

Standards, including chlorogenic acid, p-coumaric acid, scopoletin, ferulic acid, coumarin, nodakenin, bergaptol, psoralen, angelicin, 8-methoxypsoralen, bergapten, diosmetin, byakangelico, imperatorin, phelloptorin, osthole, notopterol, isoimperatorin and praeruptorin A, were purchased from the Chinese National Institute of Control of Pharmaceutical and Biological Products (Beijing, China). Thirty-two batches of samples gathered from the Sichuan province and other provinces (Gansu, Shanxi, Yunnan, Qinghai, Hebei, Shandong, Neimeng, Anhui and Shanxi provinces) were identified by Dr Yan-xu Chang, Tianjin University of Traditional Chinese Medicine. Methanol, acetonitrile and formic acid for liquid chromatography were purchased from Merck (Merck, KGaA, Darmstadt, Germany). Deionized water was purified with a Milli-Q Academic ultra-pure water system (Millipore, Milford, MA, USA) and used for the HPLC mobile phase. Other reagents were of analytical grade.

### 3.2. Preparation of Samples

The dried powdered NRR samples were weighed at 1.000 g accurately and then were ultrasonically extracted with 10 mL methanol in Erlenmeyer flasks for 30 min. Following cooling at room temperature, the samples were centrifuged for 10 min at 14,000 rpm and then the supernatant was transferred into another tube. The solutions were diluted 10 times with methanol and put into a fridge and stored at 4 °C until analysis.

Standard solutions of the 4 organic acids and 15 coumarins were weighed at 1 mg accurately and dissolved with 1 mL methanol as the stock liquid, giving a concentration of 1 mg mL^−1^. Regarding the quality control (QC) samples, the low concentration of chlorogenic acid, p-Coumaric acid, scopoletin, ferulic acid, coumarin, nodakenin, bergaptol, psoralen, angelicin, 8-Methoxypsoralen, bergapten, diosmetin, byakangelico, imperatorin, phelloptorin, osthole, notopterol, isoimperatorin and praeruptorin A were 0.3, 0.25, 0.25, 0.6, 0.4, 0.6, 0.4, 0.2, 0.2, 0.2, 0.2, 0.2, 0.4, 0.8, 0.2, 0.5, 0.6, 0.6 and 0.2 μg mL^−1^, respectively, the medium concentrations of these were 3, 2.5, 2.5, 6, 4, 6, 4, 4, 2, 2, 2, 2, 2, 4, 8, 2, 5, 6, 6 and 2 μg mL^−1^, respectively, and the high concentrations of them were 30, 25, 25, 60, 40, 60, 40, 20, 20, 20, 20, 20, 40, 80, 20, 50, 60, 60, and 20 μg mL^−1^, respectively. They were prepared by using methanol.

### 3.3. Chromatographic Conditions

A Waters Acquity UHPLC instrument (Waters Corporation, Milford, MA, USA) equipped with a photodiode array detector was used in separating the 19 active compounds and conducting qualitative analyses on them. The mobile phases consisted of a 0.1% (*v*/*v*) formic acid aqueous solution (A) and methanol and acetonitrile with ration 50%:50% (B). A gradient elution program was set at 0–12 min, 12–12% B; 12–14 min, 12–20% B; 14–17 min, 20–28% B; 17–20 min, 28–34% B; 20–22 min, 34–48% B; 22–25 min, 48–58% B; 25–32 min, 58–58% B; 32–33 min, 58–70% B; 33–35 min, 70–80% B; 35–38 min, 80–90% B; 38–39 min, 90–95% B; 39–40 min, 95–95% B. The flow rate was set at 0.3 mL min^−1^. The separation conditions were achieved on the ACQUITY UHPLC BEH C_18_ column (2.1 × 100 mm, 1.7 μm). The injection volume was set at 1 μL and the column temperature was maintained at 35 °C, while 325 nm was chosen to be the detection wavelength.

The column, column temperature, flow rate, mobile phase and gradient elution program were the same with the qualitative analysis above, while the instrument was an Agilent UHPLC–Q–TOF/MS system to identify the compounds in NRR. The ESI–MS spectra contained both negative and positive ions. The high-purity was chosen as the nebulization and auxiliary gas with the gas pressure set to 35 psig. Regarding the positive and negative ion mode, the capillary voltage was separated, set to 3.5 kV and 3.0 kV, respectively. The drying gas was set to 9.0 L min^−1^ at 350 °C, the fragmentor voltage at 125 V, skimmer voltage 65 V, and the collision energy (CE) at −30 for negative and 40 for positive mode. Data were collected at the scan range of 100–1500 Da for MS and 50–1000 Da for MS/MS. Data acquisition was analyzed by Mass Hunter software (Agilent Technologies, Santa Clara, CA, USA).

### 3.4. Calculation of Relative Response Factors 

The stock liquid of 19 compounds was used to prepare the mix standard with the different concentrations of chlorogenic acid, p-coumaric acid, scopoletin, ferulic acid, coumarin, nodakenin, bergaptol, psoralen, angelicin, 8-Methoxypsoralen, bergapten, diosmetin, byakangelico, imperatorin, phelloptorin, osthole, notopterol, isoimperatorin and praeruptorin A and were 50, 25, 25, 100, 50, 100, 50, 20, 20, 25, 25, 20, 50, 100, 25, 80, 100, 100, 25 and 50 μg mL^−1^, respectively. Then, eight concentration levels of calibration curve solution were prepared by diluting the mixed standard solution with methanol at the dilution times of 2, 5, 10, 25, 50, 125 and 250. Every compound in all the concentration levels of the calibration curve had the corresponding area in the chromatography. The external standard (ES) method was calculated by plotting the peak areas (y) against the corresponding concentration (x, μg mL^−1^), but, for the SSDM method, the relative response factors (R_f_) were calculated by the following formula:
$$R_{f}\;=\;\frac{\left(\sum_{k=1}^{n}\left(\mathrm{Ask}/\mathrm{Csk}\right)/\left(\mathrm{Aik}/\mathrm{Cik}\right)\right)}{n}$$
Csk = Ask/[R_f_ × (Aik/Cik)]

The Ask and Csk are the peak area of the standard compound and the corresponding concentration. Aik and Cik are the peak areas of the internal standard (chlorogenic acid and notopterol in this work) and the corresponding concentration.

### 3.5. Method Validation 

The method was validated including linearity, limits of quantifications (LOQs), repeatability, precision, stability and accuracy, which was both validated using the SSMD method and ES method.

The calibration curve was calculated by plotting the peak areas (y) against the corresponding concentration (x, μg mL^−1^) from each concentration. The repeatability was measured by 6 bunches of samples and expressed as the relative standard deviation (RSD). The LOD and LOQ were calculated by diluting the concentration of the mix standard in the chromatography to give a signal-to-noise (S/N) ratio of 3 and 10, respectively.

Precision was tested, including intra-day and inter-day variability. Intra-day and inter-day precision were measured by assaying the mixed standard solutions at three concentrations of QC samples (*n* = 6) within one day and three different days, respectively. Testing for stability was carried out by assaying the mix standards solution at QC sample concentrations for 0, 2, 4, 6, 8, 12 and 24 h time intervals. The recovery was tested by measuring the total concentration after adding the different amount of the original NRR at 80%, 100% and 120% of the content to the NRR, and the recovery was calculated by the following formula:Recovery (%) = 100 × (found amount − original amount)/spiked amount

### 3.6. Data Analysis

During this experiment, all analyses were carried out in triplicate and the average was determined for samples. The quantitative data of analytes were analyzed with PLS–DA and DA methods via SIMCA-P 13.0 software and SPSS 25.0 software.

## 4. Conclusions

A single standard to determine multi-components method was established for the simultaneous determination of 19 components in NRR. It was not only an economic, easy and effective method, but also a credible method when comparing to the ES method. When coupled with PLS–DA and DA, six markers were selected to be the potential chemical difference used to classify, evaluate and predict the NRR from different regions. The novel and credible method could not only solve the problem for the standards currently unavailable, but also gives us an new insight for the quantification, discrimination and classification of TCM from different regions.

## Figures and Tables

**Figure 1 molecules-24-03574-f001:**
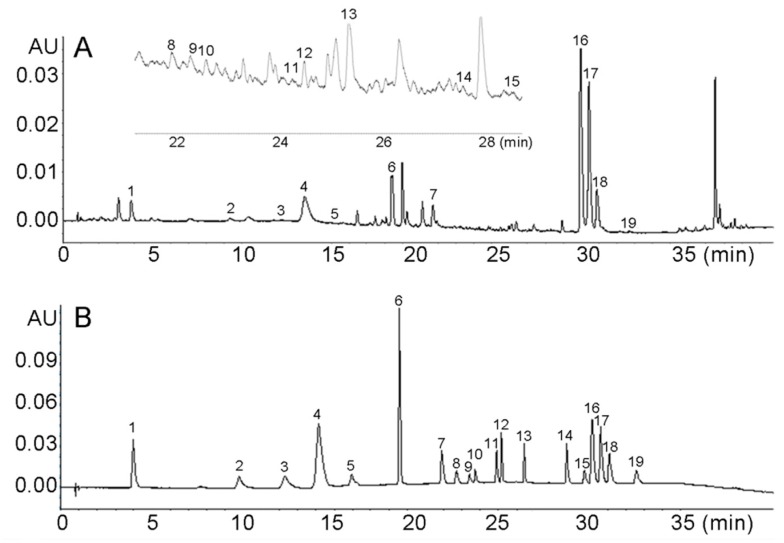
UHPLC chromatograms of *Notopterygii* rhizoma and radix sample (**A**) and standards (**B**) of 19 compounds 1: chlorogenic acid, 2: p-Coumaric acid, 3: scopoletin, 4: ferulic acid, 5: coumarin, 6: nodakenin, 7: bergaptol, 8: psoralen, 9: angelicin, 10: 8-methoxypsoralen, 11: bergapten, 12: diosmetin, 13: byakangelico, 14: imperatorin, 15: phelloptorin, 16: osthole, 17: notopterol, 18: isoimperatorin and 19: praeruptorin A.

**Figure 2 molecules-24-03574-f002:**
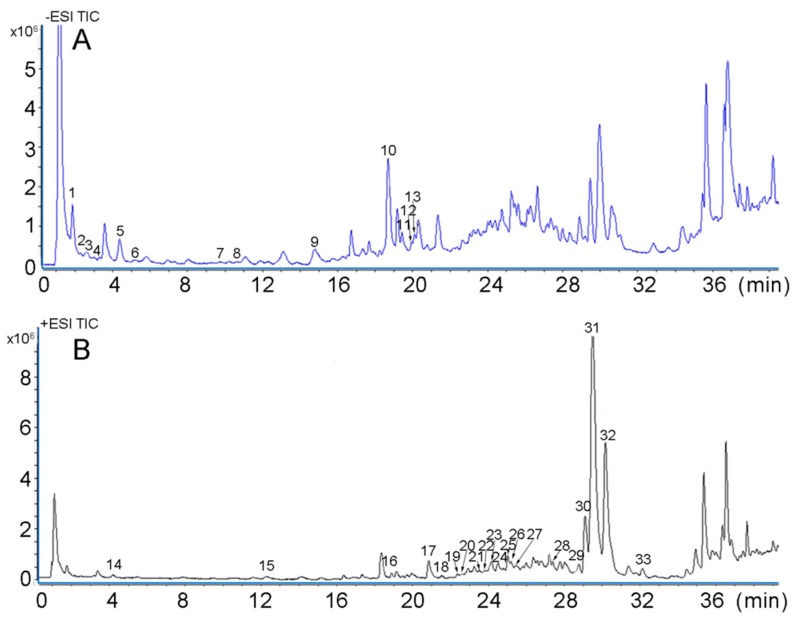
UHPLC–Q–TOF–MS/MS chromatograms of *Notopterygii* rhizoma and radix extract. (**A**) Total ion current in a negative ion mode; (**B**) Total ion current in a positive ion mode.

**Figure 3 molecules-24-03574-f003:**
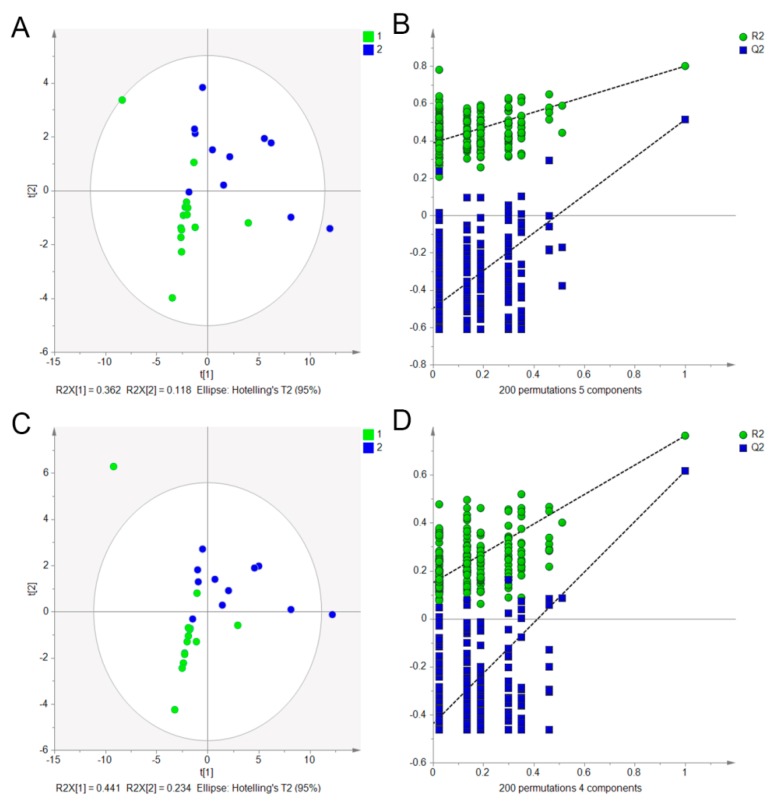
PLS–DA model for NRRs planted in the Sichuan province and others, depending on 19 compounds (**A**) with their corresponding permutation plots (**B**) and 6 markers (**C**) with the corresponding permutation plots (**D**).

**Table 1 molecules-24-03574-t001:** Data for the calibration curves, LODs, LOQs, R_f_ and Repeatability using the SSMD method (*n* = 6).

Compounds	Regressive Equation	Line Range (µg mL^−1^)	r	R_f_	LOD (µg mL^−1^)	LOQ (µg mL^−^^1^)	Repetition
							RSD(%)
Chlorogenic acid	Y = 8509.311X − 113.31	0.2–50	0.9999	1	0.03	0.1	1.25
p-Coumaric acid	Y = 7248.429X − 293.686	0.2–50	0.9996	0.783	0.06	0.2	2.39
Scopoletin	Y = 9064.661X + 291.485	0.2–50	0.9998	2.234	0.06	0.2	1.81
Ferulic acid	Y = 11890.98X − 1847.27	0.4–100	0.9999	1.291	0.06	0.2	2.36
Coumarin	Y = 2289.637X + 10.744	0.4–100	0.9999	0.528	0.15	0.4	2.69
Nodakenin	Y = 8779.59X − 291.147	0.4–100	0.9999	1.028	0.03	0.1	1.89
Bergaptol	Y = 5205.188X − 195.825	0.2–50	0.9999	1.141	0.06	0.2	2.16
Psoralen	Y = 5343.163X − 92.878	0.16–40	0.9999	1.163	0.05	0.15	4.09
Angelicin	Y = 2935.177X − 2.095	0.16–40	0.9997	0.672	0.05	0.15	3.06
8-Methoxypsoralen	Y = 2590.638X + 61.142	0.1–25	0.9999	0.628	0.06	0.2	2.86
Bergapten	Y = 5957.552X − 68.930	0.1–25	0.9999	1.329	0.03	0.1	1.63
Diosmetin	Y = 10084.27X − 186.211	0.08–20	0.9998	2.187	0.02	0.08	3.17
Byakangelico	Y = 3351.521X − 66.420	0.2–50	0.9998	0.751	0.08	0.2	1.16
Imperatorin	Y = 2247.772X − 112.236	0.4–100	0.9999	0.5	0.12	0.4	2.02
Phelloptorin	Y = 3329.094X + 50.928	0.1–25	0.9999	0.791	0.03	0.1	2.63
Osthole	Y = 8868.786X + 12.322	0.32–80	0.9999	2.035	0.03	0.1	2.61
Notopterol	Y = 4214.462X + 252.086	0.4–100	0.9999	1	0.06	0.2	1.77
Isoimperatorin	Y = 2403.511X − 29.034	0.4–100	0.9999	0.547	0.06	0.2	1.67
Praeruptorin A	Y = 5169.775X + 161.942	0.16–40	0.9999	1.272	0.05	0.15	2.63

**Table 2 molecules-24-03574-t002:** Intra-day and inter-day precision of the 19 compounds using the SSMD method (*n* = 6).

Compounds	Concentration (µg mL^−1^)	Intra-Day		Inter-Day	
		Accuracy (%)	RSD (%)	Accuracy (%)	RSD (%)
Chlorogenic acid	0.3	98.2	3.94	98.9	2.83
	3	97.1	2.02	98.0	2.20
	30	98.0	1.53	101	3.93
p-Coumaric acid	0.25	97.1	2.60	92.8	5.65
	2.5	109	2.08	110	2.69
	25	109	2.04	110	2.33
Scopoletin	0.25	109	3.24	107	3.23
	2.5	96.8	3.01	96.8	3.83
	25	95.4	3.34	95.7	2.32
Ferulic acid	0.6	85.7	2.66	84.7	2.62
	6	108	3.18	109	3.27
	60	114	1.35	114	2.01
Coumarin	0.4	105	2.80	101	3.55
	4	103	3.18	103	2.70
	40	102	3.44	102	2.89
Nodakenin	0.6	100	2.75	99.3	2.14
	6	106	1.13	106	1.33
	60	105	2.08	104	2.94
Bergaptol	0.4	92.3	2.27	95.1	3.76
	4	108	2.78	107	2.07
	40	107	1.39	105	4.14
Psoralen	0.2	92.8	2.40	96.1	3.72
	2	108	2.24	107	2.86
	20	107	2.92	109	2.94
Angelicin	0.2	100	4.67	99.9	4.57
	2	99.6	5.12	102	3.87
	20	101	3.85	101	4.34
8-Methoxypsoralen	0.2	105	1.83	106	3.23
	2	97.3	3.19	96.7	3.50
	20	97.3	2.85	96.7	3.50
Bergapten	0.2	101	2.36	99.4	3.02
	2	106	4.08	103	4.42
	20	106	2.10	102	4.21
Diosmetin	0.2	99.1	3.43	97.8	3.38
	2	110	1.40	110	2.12
	20	109	2.52	108	4.69
Byakangelico	0.4	101	4.52	98.7	3.96
	4	106	1.87	105	2.25
	40	105	2.80	106	2.87
Imperatorin	0.8	102	0.97	100	2.04
	8	108	0.95	106	1.37
	80	108	1.80	107	1.75
Phelloptorin	0.2	101	1.65	103	3.42
	2	102	3.04	99.5	3.81
	20	99.0	4.42	98.2	5.09
Osthole	0.5	101	1.12	102	2.07
	5	103	1.82	102	2.14
	50	104	1.65	104	1.93
Notopterol	0.6	108	2.24	107	2.57
	6	94.9	4.55	101	3.14
	60	91.4	2.27	94.5	2.87
Isoimperatorin	0.6	100	3.50	100	2.92
	6	104	1.66	101	3.14
	60	104	6.32	102	4.41
Praeruptorin A	0.2	109	4.58	108	3.26
	2	92.2	3.01	93.9	2.99
	20	93.8	5.45	93.8	4.45

**Table 3 molecules-24-03574-t003:** Stability of 19 compounds using the SSMD method (*n* = 6).

Compounds	Concentration (µg mL^−1^)	Stability	
		Accuracy (%)	RSD (%)
Chlorogenic acid	0.3	96.6	4.98
	3	96.8	2.64
	30	95.4	1.70
p-Coumaric acid	0.25	99.1	3.12
	2.5	107	2.94
	25	113	5.66
Scopoletin	0.25	103	5.46
	2.5	96.5	4.25
	25	99.1	3.36
Ferulic acid	0.6	83.2	4.54
	6	114	2.84
	60	121	5.43
Coumarin	0.4	102	5.05
	4	104	5.47
	40	108	3.32
Nodakenin	0.6	97.8	2.92
	6	108	3.61
	60	108	3.13
Bergaptol	0.4	98.5	1.85
	4	107	3.22
	40	113	3.29
Psoralen	0.2	98.0	4.82
	2	108	3.56
	20	112	5.07
Angelicin	0.2	96.6	4.67
	2	104	4.13
	20	104	5.21
8-Methoxypsoralen	0.2	103	3.96
	2	102	5.87
	20	101	3.79
Bergapten	0.2	100	4.99
	2	111	2.79
	20	111	5.03
Diosmetin	0.2	101	4.32
	2	113	5.68
	20	119	3.76
Byakangelico	0.4	99.3	6.31
	4	111	3.28
	40	107	2.94
Imperatorin	0.8	98.8	3.80
	8	110	4.29
	80	107	1.29
Phelloptorin	0.2	100	4.21
	2	99.1	2.93
	20	103	1.58
Osthole	0.5	103	2.68
	5	107	5.18
	50	103	4.83
Notopterol	0.6	104	1.71
	6	93.7	6.51
	60	91.5	3.22
Isoimperatorin	0.6	99.0	3.04
	6	101	3.77
	60	106	5.44
Praeruptorin A	0.2	103	4.30
	2	94.2	3.53
	20	92.1	3.32

**Table 4 molecules-24-03574-t004:** Recovery of 19 compounds using the SSMD method (*n* = 6).

Compounds	Origin	Added	SSMD	
			Recovery (%)	RSD (%)
Chlorogenic acid	3.47	2.80	102	3.00
		3.50	109	4.37
		4.20	98.5	3.26
p-Coumaric acid	1.09	0.80	122	2.62
		1.00	124	3.30
		1.20	119	6.29
Scopoletin	0.22	0.16	97.3	4.31
		0.20	95.2	2.06
		0.24	96.6	3.64
Ferulic acid	7.97	6.40	123	5.35
		8.00	122	5.49
		9.60	122	1.99
Coumarin	0.31	0.24	100	3.95
		0.30	104	7.59
		0.36	93.0	2.80
Nodakenin	7.93	6.40	107	3.01
		8.00	113	6.18
		9.60	108	4.45
Bergaptol	3.95	3.20	110	4.23
		4.00	109	7.53
		4.80	100	4.95
Psoralen	0.17	0.16	102	6.19
		0.20	97.3	7.02
		0.24	101	3.05
Angelicin	0.37	0.32	96.2	5.49
		0.40	102	2.64
		0.48	94.2	8.10
8-Methoxypsoralen	0.40	0.32	90.8	1.23
		0.40	95.9	7.64
		0.48	87.9	4.55
Bergapten	0.20	0.16	101	5.76
		0.20	108	4.56
		0.24	106	5.85
Diosmetin	0.04	0.03	52.0	9.25
		0.04	60.6	3.96
		0.05	65.5	8.84
Byakangelico	2.68	2.40	105	6.82
		3.00	108	4.64
		3.60	106	7.47
Imperatorin	0.30	0.24	85.0	6.08
		0.30	85.9	0.75
		0.36	82.9	7.83
Phelloptorin	0.13	0.12	104	3.85
		0.15	112	5.98
		0.18	102	5.38
Osthole	27.40	24.00	104	2.43
		30.00	109	5.11
		36.00	91.1	7.64
Notopterol	48.42	40.00	85.6	6.93
		50.00	89.0	3.51
		60.00	89.7	2.75
Isoimperatorin	22.93	16.00	104	4.14
		20.00	107	5.69
		24.00	106	6.42
Praeruptorin A	0.32	0.24	96.8	6.48
		0.30	103	6.90
		0.40	103.0	3.96

**Table 5 molecules-24-03574-t005:** UHPLC–Q–TOF data and identification of constituents from *Notopterygii* rhizoma and radix.

Peak No.	Rt (min)	*m/z*	M	Formula	Fragmentation	ppm	Identification
1	1.5	115.0034	[M − H]^−^	C_4_H_4_O_4_	99.9242,73.0302,53.0038	2.43	fumaric acid
2	2.3	353.0893	[M − H]^−^	C_16_H_18_O_9_	233.0436, 205.0487,191.0553,127.0327	−4.22	neochlorogenic acid
3	2.6	341.0885	[M − H]^−^	C_15_H_18_O_9_	179.0347,135.044	−2.03	Caffeic acid 3-glucoside
4	2.7	153.0186	[M − H]^−^	C_7_H_6_O_4_	109.0287,91.0178	4.75	2,5-Dihydroxybenzoic acid
5	4.3	353.0893	[M − H]^−^	C_16_H_18_O_9_	191.0559,205.0487,135.045	−4.22	chlorogenic acid
6	5.1	353.0882	[M − H]^−^	C_16_H_18_O_9_	191.0559,205.0487	−1.11	cryptochlorogenin acid
7	9.8	515.1195	[M − H]^−^	C_25_H_24_O_12_	353.0916,191.0558,179.0349,161.0251,135.045	0.00	isochlorogenic acid B
8	10.7	163.0397	[M − H]^−^	C_9_H_8_O_3_	119.0495,93.034	2.24	p-Coumaric acid
9	14.7	193.0507	[M − H]^−^	C_10_H_10_O_4_	135.0411,133.0287,89.0324	−0.35	ferulic acid
10	18.8	453.1405	[M − H]^−^	C_20_H_24_O_9_	227.0703,89.0238	−0.65	nodakenin
11	19.2	515.1206	[M − H]^−^	C_25_H_24_O_12_	353.0916,191.0558,179.0349,161.0251,135.045	-2.13	isochlorogenic acid A
12	20.3	515.1196	[M − H]^−^	C_25_H_24_O_12_	353.0916,191.0558,179.0349,161.0251,135.046	−0.19	isochlorogenic acid C
13	20.5	607.1672	[M − H]^−^	C_28_H_32_O_15_	300.0596,284.0330	−0.59	diosmin-d3
14	4.2	163.0389	[M + H]^+^	C_9_H_6_O_3_	145.0482	0.44	umbelliferone
15	12.3	193.0492	[M + H]^+^	C_10_H_8_O_4_	94.0412,178.0243,150.0289,122.0363,66.0465	1.75	Scopoletin
16	18.2	207.0642	[M + H]^+^	C_11_H_10_O_4_	146.0484	4.78	6,7-Dimethoxy coumarin
17	21.2	203.0348	[M + H]^+^	C_11_H_6_O_4_	147.0443,119.0502,91.0541,65.0387	−4.53	bergaptol
18	21.9	187.039	[M + H]^+^	C_11_H_6_O_3_	131.0489	−0.16	psoralen
19	22.2	187.0401	[M + H]^+^	C_11_H_6_O_3_	131.0490	−4.17	angelicin
20	22.3	305.1025	[M + H]^+^	C_16_H_16_O_6_	203.0331,147.0433,131.0483,159.0438	−1.76	oxypeucedanin hydrate
21	23.1	217.051	[M + H]^+^	C_12_H_8_O_4_	146.0453,161.0601	−6.78	8-Methoxypsoralen
22	24	247.0602	[M + H]^+^	C_13_H_10_O_5_	217.0112,189.0178,161.0230,133.0280	−0.41	pimpinellin
23	24.1	217.0472	[M + H]^+^	C_12_H_8_O_4_	174.0305,146.0355,118.0408,131.0494,202.0249	−0.76	bergapten
24	24.2	301.0721	[M + H]^+^	C_16_H_12_O_6_	217.0134,173.0284	−5.12	diosmetin
25	24.8	247.0602	[M + H]^+^	C_13_H_10_O_5_	217.0112,189.0178,161.0230,133.0280	−0.41	isopimpinellin
26	25	203.0339	[M + H]^+^	C_11_H_6_O_4_	119.0846,105.0698,91.0544	−0.07	xanthotol
27	25.2	231.1016	[M + H]^+^	C_14_H_14_O_3_	147.0434,119.0488	−0.13	7-(3-methylbut-2-enoxy)chromen-2-one
28	27.7	271.0969	[M + H]^+^	C_16_H_14_O_4_	203.0328,147.0433,91.0545	−1.54	imperatorin
29	28.0	301.1092	[M + H]^+^	C_17_H_16_O_5_	217.0144,189.0173,173.0281,218.0197,162.0312	−7.16	phelloptorin
30	28.5	245.1190	[M + H]^+^	C_15_H_16_O_3_	189.1097,131.0491	−2.19	osthole
31	29.1	377.1369	[M + Na]^+^	C_21_H_22_O_5_	203.0345,147.0412,91.0529,159.0391,131.0506	−2.7	notopterol
32	30.1	271.0976	[M + H]^+^	C_16_H_14_O_4_	203.0329,147.0437,91.0545,159.0433,131.0486	−4.31	isoimperatorin
33	31.6	409.1277	[M + H]^+^	C_21_H_22_O_7_	409.1277	−4.99	praeruptorin A

**Table 6 molecules-24-03574-t006:** Contents of 19 compounds in different samples using the SSMD method and the result of discriminant analysis.

Constant (mg g^−1^) SSMD	Compounds																				
	Chlorogenic Acid	p-Coumaric Acid	Scopoletin	Ferulic Acid	Coumarin	Nodakenin	Bergaptol	Psoralen	Angelicin	8-Methoxypsoralen	Bergapten	Diosmetin	Byakangelico	Imperatorin	Phelloptorin	Osthole	Notopterol	Isoimperatorin	Praeruptorin A	Discrimination	Score
1Sichuan(chengdu)	0.55	0.20	0.00	0.75	0.02	1.82	0.15	0.01	0.02	0.02	0.01	0.00	0.49	0.00	0.12	4.53	2.46	1.44	0.12	IR	
2Sishuan(meishan)	3.05	0.75	0.03	2.53	0.03	1.56	0.22	0.08	0.06	0.09	0.06	0.00	1.25	0.05	0.04	7.08	8.92	4.33	0.00	1	3.562
3Sichuan(leshan)	0.34	0.10	0.02	0.80	0.00	0.83	0.40	0.02	0.02	0.06	0.03	0.00	0.24	0.05	0.03	2.62	4.42	2.25	0.03	1	0.556
4Sichuan(ya’an)	0.26	0.22	0.03	0.66	0.00	0.75	0.13	0.02	0.06	0.04	0.01	0.30	0.43	0.37	0.14	2.80	4.89	3.41	0.03	1	0.966
5Sichuan(guanghan)	0.79	0.08	0.00	0.87	0.07	0.96	0.12	0.00	0.03	0.07	0.16	0.02	0.67	104.51	0.04	0.30	1.17	22.81	0.44	1	2.087
6Sichuan(mianyang)	0.77	0.14	0.03	1.15	0.02	5.63	0.33	0.04	0.09	0.10	0.17	3.12	1.39	2.53	0.21	2.17	3.71	4.51	0.00	1	2.304
7Sichuan(luzhou)	0.24	0.31	0.02	0.81	0.02	1.36	0.33	0.03	0.02	0.06	0.03	0.02	0.50	0.32	0.04	4.81	5.99	3.99	0.02	1	1.912
8Shanxi	0.15	0.56	0.00	0.03	0.00	0.16	0.15	0.04	0.49	0.59	0.04	0.00	0.10	0.20	0.10	0.16	3.67	0.54	1.38	2	−1.599
9Gansu	0.71	0.17	0.03	0.86	0.02	1.24	0.21	0.04	0.02	0.06	0.04	0.02	0.43	0.10	0.03	2.26	3.66	2.16	0.03	1	0.227
10Gansu	1.08	0.05	0.08	0.66	0.04	29.73	1.53	0.03	0.20	0.15	0.14	0.03	3.18	0.12	0.61	0.06	0.19	31.07	0.00	2	−2.789
11Sichuan(yibin)	1.76	0.53	0.04	1.21	0.01	0.78	0.15	0.05	0.13	0.11	0.02	0.00	0.73	0.11	0.03	4.50	5.04	1.98	0.00	1	1.586
12Sichuan(xinjin)	0.10	0.03	0.00	0.29	0.01	0.36	0.63	0.02	0.01	0.07	0.02	0.28	0.07	0.14	0.04	0.59	2.25	3.29	0.00	1	−0.602
13Sichuan(bazhong)	0.17	0.04	0.00	0.49	0.03	23.17	2.91	0.01	0.13	0.13	0.15	5.96	3.12	0.34	0.57	0.44	0.82	22.57	0.06	1	2.009
14Sichuan(neijiang)	0.34	0.15	0.00	0.88	0.00	0.70	0.09	0.04	0.02	0.10	0.02	0.02	0.61	0.02	0.03	3.20	5.63	1.21	0.24	1	0.967
15Sichuan(ziyang)	1.28	0.44	0.03	1.08	0.01	0.81	0.34	0.03	0.06	0.09	0.03	0.00	0.69	0.22	0.03	4.01	5.86	2.26	0.00	1	1.531
16Yunnan(kunming)	1.59	0.14	0.03	0.61	0.01	28.10	0.81	0.03	0.27	0.12	0.28	0.28	2.23	0.51	0.37	0.05	0.19	26.10	0.00	2	−2.528
17Qinhai	2.75	0.25	0.03	0.93	0.01	16.61	0.22	0.03	0.16	0.07	0.22	0.01	0.33	0.00	0.96	1.10	1.76	10.29	0.00	2	−1.399
18Sichuan	0.88	0.23	0.05	0.88	0.02	5.06	0.62	0.08	0.04	0.22	0.03	0.01	0.28	4.71	0.09	6.23	0.85	3.93	0.36	1	1.264
19Sichuan	0.32	0.22	0.04	0.86	0.01	4.43	1.72	0.02	0.04	0.11	0.00	0.03	0.36	6.96	0.06	4.55	2.69	4.88	0.00	1	1.208
20Sichuan	0.33	0.31	0.00	0.00	0.00	0.10	0.31	0.03	0.02	0.21	0.01	0.00	0.03	0.02	0.00	0.14	0.54	2.83	0.73	IR	
21Sichuan	0.37	0.14	0.05	0.81	0.01	1.36	0.79	0.07	0.04	0.16	0.03	0.25	0.34	0.41	0.04	3.18	0.59	2.14	0.00	IR	
22Shandong(taishan)	1.52	0.26	0.03	1.12	0.07	1.23	0.17	0.07	0.06	0.15	0.06	0.00	0.28	0.45	0.02	2.35	1.94	0.89	0.83	IR	
23Neimeng	0.17	1.04	0.03	0.09	0.03	0.14	0.16	0.05	1.71	0.26	0.06	0.01	0.10	0.40	0.32	0.16	1.39	3.04	2.78	2	−2.897
24Gansu	0.31	0.06	0.02	0.55	0.03	8.56	0.38	0.00	0.08	0.07	0.04	0.00	0.46	0.07	0.06	0.65	0.79	6.26	0.00	2	−1.451
25Gansu	0.80	0.11	0.00	1.19	0.01	15.06	0.81	0.03	0.20	0.08	0.09	0.03	1.54	0.00	0.13	2.06	2.29	12.18	0.00	2	−0.756
26Anhui	0.15	0.12	0.02	0.41	0.01	0.46	0.43	0.05	0.01	0.51	0.06	0.01	0.17	0.61	0.02	1.86	4.12	2.63	4.35	2	−2.526
27Sichuan	0.29	0.75	0.00	2.37	0.04	1.04	0.45	0.03	0.02	0.06	0.05	0.00	0.64	0.00	0.04	4.67	6.80	11.25	0.00	1	2.017
28Sichuan	0.11	0.06	0.00	0.51	0.01	0.56	0.23	0.00	0.01	0.02	0.04	0.01	0.14	0.09	0.00	1.39	0.71	0.85	0.02	IR	
29Tianjin	0.19	0.41	0.02	0.61	0.01	0.60	0.17	0.03	0.02	0.08	0.00	0.01	0.56	0.03	0.03	3.37	0.52	0.84	0.00	IR	
30Shanxi	0.49	0.06	0.00	1.28	0.00	32.45	3.06	0.02	0.13	0.15	0.31	0.03	0.54	0.28	0.52	0.04	3.16	0.04	0.00	IR	
31Shanxi	1.68	0.22	0.00	1.06	0.08	49.95	3.10	0.03	0.25	0.14	0.30	0.06	3.30	0.41	0.66	0.01	6.16	0.02	0.00	2	−2.453
32Shanxi	0.74	0.15	0.00	1.47	0.00	71.70	3.93	0.03	0.30	0.34	0.25	0.05	2.24	0.64	0.62	0.01	7.13	0.06	0.00	2	−3.236

Note: IR means irregular sample according Chinese Pharmacopoeia (2015).

**Table 7 molecules-24-03574-t007:** The RD value from comparing the ES with the SSMD method.

RD (%)	Compounds																		
	Chlorogenic Acid	p-Coumaric acid	Scopoletin	Ferulic Acid	Coumarin	Nodakenin	Bergaptol	Psoralen	Angelicin	8-Methoxypsoralen	Bergapten	Diosmetin	Byakangelico	Imperatorin	Phelloptorin	Osthole	Notopterol	Isoimperatorin	Praeruptorin A
1Sichuan(chengdu)	1.34	8.09	0.00	7.53	1.51	1.79	1.98	6.42	0.11	4.43	8.52	0.00	1.95	0.00	2.26	0.10	3.15	0.63	4.55
2Sishuan(meishan)	1.54	9.82	2.58	9.24	0.82	1.76	2.79	3.08	0.09	2.91	0.88	0.00	2.21	7.56	0.01	0.10	3.32	0.68	0.00
3Sichuan(leshan)	1.18	5.56	7.85	7.69	4.74	1.56	3.59	2.65	0.13	1.36	0.81	0.00	1.52	6.69	1.89	0.10	3.25	0.66	1.70
4Sichuan(ya’an)	1.05	8.29	1.86	7.21	3.12	1.52	1.55	2.73	0.09	0.25	6.65	5.03	1.90	1.61	2.46	0.10	3.27	0.68	5.42
5Sichuan(guanghan)	1.41	4.81	0.00	7.87	0.15	1.62	1.33	0.00	0.00	2.31	2.04	5.23	2.07	3.00	0.04	0.06	2.90	0.71	6.21
6Sichuan(mianyang)	1.40	7.04	4.28	8.37	1.78	1.92	3.37	0.92	0.13	3.15	2.09	5.64	2.22	2.80	2.76	0.10	3.23	0.68	0.00
7Sichuan(luzhou)	1.02	8.91	8.55	7.71	1.68	1.73	3.37	0.33	0.20	1.23	1.58	4.33	1.96	1.37	0.52	0.10	3.29	0.68	11.5
8Shanxi	0.70	9.60	0.00	7.93	2.92	0.18	1.99	0.58	0.19	4.87	0.13	0.00	0.25	0.48	2.07	0.02	3.23	0.49	6.63
9Gansu	1.39	7.69	3.24	7.85	2.10	1.70	2.66	0.19	0.09	1.78	0.49	5.85	1.89	1.85	1.02	0.10	3.23	0.65	3.34
10Gansu	1.46	1.32	3.37	7.21	0.55	1.97	4.34	0.09	0.17	3.77	1.92	0.06	2.31	1.33	3.22	0.13	0.27	0.71	0.00
11Sichuan(yibin)	1.50	9.55	0.46	8.45	3.72	1.54	1.87	1.71	0.15	3.21	2.31	0.00	2.09	1.77	0.89	0.10	3.27	0.65	0.00
12Sichuan(xinjin)	0.27	5.56	0.00	3.87	4.11	1.03	3.95	3.48	1.19	2.10	2.10	4.98	0.51	0.60	0.35	0.08	3.13	0.67	0.00
13Sichuan(bazhong)	0.79	1.50	0.00	6.26	1.15	1.96	4.46	8.53	0.15	3.60	2.00	5.68	2.31	1.49	3.20	0.07	2.69	0.71	2.15
14Sichuan(neijiang)	1.18	7.26	0.00	7.89	0.00	1.48	0.21	0.48	0.25	3.07	1.92	5.34	2.03	14.9	1.06	0.10	3.28	0.61	5.67
15Sichuan(ziyang)	1.48	9.35	2.12	8.27	3.36	1.55	3.40	0.03	0.08	2.72	1.33	0.00	2.07	0.70	2.37	0.10	3.29	0.66	0.00
16Yunnan(kunming)	1.50	7.05	5.04	6.99	3.24	1.97	4.10	1.08	0.18	3.40	2.35	4.98	2.28	1.98	3.07	0.19	0.37	0.71	0.00
17Qinhai	1.53	8.51	4.61	7.99	2.77	1.96	2.74	1.58	0.16	1.86	2.24	9.19	1.75	0.00	3.31	0.09	3.06	0.70	0.00
18Sichuan	1.43	8.41	1.05	7.89	1.66	1.91	3.94	3.05	0.04	4.26	0.64	8.12	1.65	2.90	1.89	0.10	2.72	0.68	6.07
19Sichuan	1.15	8.31	0.14	7.84	4.02	1.90	4.37	5.46	0.03	3.18	0.00	0.78	1.80	2.93	0.97	0.10	3.17	0.69	0.00
20Sichuan	1.17	8.93	0.00	0.00	0.00	1.45	3.31	0.83	0.14	4.20	5.78	0.00	4.30	14.6	0.00	0.00	2.33	0.67	6.46
21Sichuan	1.22	7.08	0.88	7.72	6.91	1.73	4.08	2.75	0.02	3.85	1.74	4.90	1.77	1.73	0.60	0.10	2.43	0.65	0.00
22Shandong(taishan)	1.49	8.63	5.39	8.33	0.13	1.70	2.28	2.60	0.08	3.82	0.65	0.00	1.64	1.85	6.03	0.10	3.09	0.57	6.50
23Neimeng	0.76	10.0	4.01	7.60	1.08	0.45	2.13	1.31	0.20	4.40	0.66	9.98	0.39	1.72	3.01	0.02	2.98	0.67	6.73
24Gansu	1.15	2.38	7.27	6.64	1.12	1.94	3.53	0.00	0.12	1.97	0.54	0.00	1.92	3.82	0.90	0.08	2.67	0.69	0.00
25Gansu	1.41	6.26	0.00	8.42	4.82	1.96	4.10	1.32	0.17	2.41	1.48	0.86	2.24	0.00	2.37	0.09	3.14	0.70	0.00
26Anhui	0.65	6.61	9.07	5.55	6.25	1.23	3.65	1.92	0.88	4.81	0.84	10.5	1.18	2.15	3.21	0.09	3.25	0.67	6.77
27Sichuan	1.12	9.81	0.00	9.19	0.69	1.65	3.70	0.73	0.14	1.66	0.17	0.00	2.05	0.00	0.12	0.10	3.30	0.70	0.00
28Sichuan	0.40	3.04	0.00	6.41	5.84	1.36	2.88	0.00	0.43	7.53	0.44	14.4	0.94	2.36	0.00	0.09	2.59	0.57	5.53
29Tianjin	0.85	9.29	9.33	6.99	4.56	1.40	2.25	1.02	0.21	2.46	0.00	11.4	2.00	12.7	1.08	0.10	2.30	0.57	0.00
30Shanxi	1.30	2.74	0.00	8.53	0.00	1.97	4.47	3.15	0.16	3.85	2.40	1.64	1.99	1.13	3.18	0.25	3.20	2.07	0.00
31Shanxi	1.50	8.27	0.00	8.24	0.06	1.97	4.47	1.81	0.18	3.73	2.38	2.39	2.31	1.73	3.24	2.19	3.29	4.11	0.00
32Shanxi	1.40	7.37	0.00	8.72	9.90	1.97	4.50	0.51	0.19	4.61	2.29	1.69	2.28	2.19	3.22	2.70	3.30	1.41	0.00

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
