# Peer review of "A Single Standard to Determine Multi-Components Method Coupled with Chemometric Methods for the Quantification, Evaluation and Classification of Notopterygii Rhizoma et Radix from Different Regions"

_molecules, 2019, doi:10.3390/molecules24193574_

Round 1

Reviewer 1 Report

This paper describes an UHPLC-Q-TOF/MS study to determine the bio-active components of Notopterygii Rhizoma et Radix (NRRNRR) and attempt a discrimination of NRR samples cultivated in different regions. The scientific quality of the manuscript is acceptable as concerning the experimental part (chromatographic analysis, MS identification and method validation). Nevertheless, chemometric treatment of the experimental data is rather difficult to understand and questionable. If follows that the declared scope of NRR geographical discrimination was completely missed. In addition, multivariate optimisation of the UHPLC-Q-TOF/MS method was not described at all. In summary, the multivariate statistical data treatment is not adequate.

I have listed below the major critical points:

-Paragraph 2.1. Optimization of extraction conditions. The authors declared that they used an experimental design (orthogonal L9) to optimize the effect of some experimental variables on the chromatogram intensity. However, they did not provide any information concerning the statistical treatment of the experimental data, goodness of the statistical model, and so on.

-Paragraph 2.2. Optimization of chromatographic conditions. It was not reported how optimization was carried out. Did the authors apply a uni-variate approach or some multivariate statistical strategies?

-Paragraph 2.7. Principal component analysis and discriminant analysis of all the samples. Apparently, all the results of geographical discrimination by principal component analysis (PCA) and discriminant analysis (DA) is summarized by Figures 3A and 3B. According to the figure captions, Figure 3A  and Figure 3B represent PLS (partial least-square)  models built with two (???!!!) or 19 variables, respectively. It follows that no PCA results are shown in Figure 3 and nowhere in the manuscript. PLS and PCA are different strategies. Actually, I am not able to understand the meaning of Figures 3A and 3B because indications of the quantities represented by the axis  were not given. It seems, as also reported in the text, that multivariate statistical analysis presented in Figure 3 A is based on two molecules. In this condition, multivariate (PCA, PLS or DA) is absolutely meaningless and useless. Reporting the samples on the variable1-variable 2 plane is enough to reveal possible clustering according to the geographical origin of samples. It seems to me that the authors made some confusion about the various statistical approaches. Figure 3  seems to represent PLS results, but in the text a linear delimiter between the classes is given, which is typical of a linear discriminant analysis approach. Which of these two strategies was really applied? Apparently, linear discriminant analysis based on 19 variables was applied to discriminate 25 samples into two classes. It follows that the sample to variable ratio is close to unity. Linear discriminant analysis cannot applied under this condition.

Reviewer 2 Report

Review

The work entitled “A single standard to determine multi-components method coupled with discriminant analysis and principal component analysis for the quantification, evaluation and classification of Notopterygii Rhizoma et Radix from different regions” described in the manuscript could be suitable for publication on Molecules after major revisions.

The relevancy of the research is well contextualized. Nevertheless, in my opinion, few modifications are needed, in order to clarify the procedures followed to calculate the models, and to increase the general readability of the paper.

Q1) Validation: In chemometrics, it is important to validate models, in order to understand how reliable results are. In general, it is possible to follow two approaches: the internal and the external validation. External validation is preferable, but, in the present case, due to the number of available samples, is probably impossible to carry it out.  

I think how models are validatated should be better described.  At Line 233 you mentioned cross-validation but you didn’t provide any detail about it (for instance, what’s the number of cancellation groups?).

Q1.1) How is model complexity (i.e., the number of principal components/PLS-scores) defined? How did you choose the number of components to be extracted?

Q2) Preprocessing: Prior the creation of chemometric models it is important to pretreat data, in order to remove spurious variability, and to mean center the signals.

Which preprocessing approaches have you applied on data? Have you mean centered or auto-scaled the signals prior the creation of the models?

Q3) Title: In my opinion the title does not well represent the aim of the work. I would suggest to the authors to give a new title, more focused and maybe a bit shorter.

Q4) Introduction: In my opinion, the introduction is well structured. The relevancy of the study is widely exposed and supported by literature. The state of the art is well discussed, providing a good overview on the analysis of NRR. Nevertheless, the statistical/chemometric tools used for data analysis are not mentioned. Several studies are present into the literature discussing chemometrics in agro-food contexts; for instance, an overview of chemometric approaches for the analysis of plants is provided in:

Biancolillo, F. Marini, Chapter Four - Chemometrics Applied to Plant Spectral Analysis, in: J. Lopes, C. Sousa (Eds.), Vibrational Spectroscopy for Plant Varieties and Cultivars Characterization, Comprehensive Analytical Chemistry, 80, Elsevier, Amsterdam, 2018, pp. 69-104.

Additionally, into the literature is easy to find papers were statistical tools are used to investigate Chinese herbal medicines. For instance, one example where the analytical/statistical techniques are close to those proposed by the authors is: 

Huang, W.-P., Tan, T., Li, Z.-F., OuYang, H., Xu, X., Zhou, B., Feng, Y.-L. Structural characterization and discrimination of Chimonanthus nitens Oliv. leaf from different geographical origins based on multiple chromatographic analysis combined with chemometric methods (2018) Journal of Pharmaceutical and Biomedical Analysis, 154, pp. 236-244.

Where UHPLC-QTOF-MS/MS analysis is coupled with chemometrics (PCA) to investigate a plant exloited in Traditional Chinese Medicine.

Consequently, I would suggest the authors to briefly discuss these aspects in the introduction and to refer to these suggested papers and, if possible, also to other present into the literature.

Q5) Section 2.1 Line 74: The authors wrote: “In order to obtain the higher amounts percentage of the 19 active compounds”; to me, this sentence is not very clear. What do you mean with: “to obtain the higher amounts percentage”? To optimize the extraction? Please, re-phrase this sentence.

Q6) Table 1: The column entitled “weight” does not present any information. If it is a typo, please correct it; otherwise, please remove this column.

Q7) Section 2.7. In my opinion, understanding the details of the chemometric analysis is a bit difficult. Consequently, I would suggest to expand this section, providing more details. In particular:

Q7.1) Lines 298-209. To me, it is not clear what is the starting point of the chemometric analysis. Could you please re-phrase these two lines, explaining on which data are you calculating the PCA?
It would be very useful if you could even report dimension of the data matrix. 

Please, address the reader to literature on PCA. One suitable reference would be:

I.T. Jolliffe, Principal Component Analysis, second edition, Springer: New York, NY, 2002.

Q7.2) Lines 209. At this line authors mention PLS. PLS is a regression method, and it can’t be used for classification. Maybe it has been coupled with a classifier, or it is PLS-DA. Please, carefully check which classifier you used and cite the original paper where it has been proposed. 

Q7.3) Line 335: At this line, the authors mention a “discriminant analysis method”. Several discriminant approaches are present into the literature, please provide name and references for the one you used.

Q8) Section 3.2. Line 266-267

Please, re-phrase this sentence.

Q9) Figure 3) Please clarify the axis labels; I guess they are components/scores. Please state somewhere what R stands for.

Q9.1) The number of samples reported in Section 2.6 is 32. Are the samples displayed in Fig.3 32? They seem less than 32.  

Round 2

Reviewer 1 Report

The authors should explain in details how the experimental data collected by the orthogonal L9 design were treated to find the optimal conditions (lines 87-91). Did they use a surface response methodology or some other statistical treatment?

Reviewer 2 Report

In agreement with the reviewers’ comment, the authors modified the manuscript. So far, several necessary details have been added to the updated version of the manuscript, representing a general improvement of the paper; nevertheless, some aspects still need to be discussed and clarified in the text.

Q1R2: In the comments to the original submission, I have asked the authors the following question (Q1.1):  How is model complexity (i.e., the number of principal components/PLS-scores) defined? How did you choose the number of components to be extracted?”

The response is:
Response 1.1: Thank you for your question. As we all know, the genuine producing area of NRR was planted in Sichuan province, called ‘Chuan qianghuo’, which has been highly valued as the superior quality. In order to explore the chemical difference of NRR planted in Sichuan province and others, a single standard to determine multi-components method was established for the simultaneous determination of 19 compounds in of 32 batches of samples planted in different regions. According to the Chinese Pharmacopoeia (2015) rules that the content of notopterol and isoimperatorin in NRR should more than 0.4%. Hence, there were 25 batches. To screen the potential compounds that caused the chemical difference between NRRs planted in Sichuan province and others, 25 batches samples were subjected to PLS-DA to validate whether they could be grouped into two parts according to the 19 coumpounds. Depending on the VIP value of PLS-DA, six compounds were selected to be the potential chemical markers for the discrimination of Notopterygii Rhizoma et Radix planted in different regions.

I thank the authors for providing me these details, nevertheless, they did not clearly address my question. What I would like to know is: How many latent variables did you extract building the PLS-DA model?

Additionally, how did the authors define the number of these components?

Please, provide this information also in the manuscript.

Q2R2: In my comments about the submission, I have asked the authors some details about the preprocessing approaches. Here is my comment (Q2):

“Preprocessing: Prior the creation of chemometric models it is important to pretreat data, in order to remove spurious variability, and to mean center the signals.

Which preprocessing approaches have you applied on data? Have you mean centered or auto-scaled the signals prior the creation of the models?”

Reply:

“ Response 2: Thank you for your advice. In our experiment, the datas for PLS-DA and DA were the absolute quantitative data of 19 compounds, which were expressed as mg/g meaning that the detial content of 19 compounds in 1 g of Notopterygii Rhizoma et Radix. And the DA model was also used to predict Notopterygii Rhizoma et Radix from the different regions depending on the absolute quantitative data of the compounds. Hence, the datas were suitable for the chemometric models.”

Also in this case the authors kindly provided me further details, but they did not reply to my question.
Did you center data prior the creation of the models? Please, state it also in the manuscript.

Q3R2:

Here is the fourth question I asked to the authors:

“Q4) Introduction: In my opinion, the introduction is well structured. The relevancy of the study is widely exposed and supported by literature. The state of the art is well discussed, providing a good overview on the analysis of NRR. Nevertheless, the statistical/chemometric tools used for data analysis are not mentioned. Several studies are present into the literature discussing chemometrics in agro-food contexts; for instance, an overview of chemometric approaches for the analysis of plants is provided in:

Biancolillo, F. Marini, Chapter Four - Chemometrics Applied to Plant Spectral Analysis, in: J. Lopes, C. Sousa (Eds.), Vibrational Spectroscopy for Plant Varieties and Cultivars Characterization, Comprehensive Analytical Chemistry, 80, Elsevier, Amsterdam, 2018, pp. 69-104.

Additionally, into the literature is easy to find papers were statistical tools are used to investigate Chinese herbal medicines. For instance, one example where the analytical/statistical techniques are close to those proposed by the authors is: Huang, W.-P., Tan, T., Li, Z.-F., OuYang, H., Xu, X., Zhou, B., Feng, Y.-L. Structural characterization and discrimination of Chimonanthus nitens Oliv. leaf from different geographical origins based on multiple chromatographic analysis combined with chemometric methods (2018) Journal of Pharmaceutical and Biomedical Analysis, 154, pp. 236-244.Where UHPLC-QTOF-MS/MS analysis is coupled with chemometrics (PCA) to investigate a plant exloited in Traditional Chinese Medicine. Consequently, I would suggest the authors to briefly discuss these aspects in the introduction and to refer to these suggested papers and, if possible, also to other present into the literature.”

I have suggested the authors to enlarge the rationale behind the chemometric part of the work. I have suggested them two works to be cited and I have asked to add even more by themselves. Nevertheless, they did not put a large effort on this, in fact, they only added one paper. Consequently, I would encourage the authors to take more care of this suggestion.
